# Effects of Different Moisture Levels and Additives on the Ensiling Characteristics and In Vitro Digestibility of *Stylosanthes* Silage

**DOI:** 10.3390/ani12121555

**Published:** 2022-06-16

**Authors:** Jinze Bao, Lei Wang, Zhu Yu

**Affiliations:** College of Grassland Science and Technology, China Agricultural University, Beijing 100193, China; bs20203240986@cau.edu.cn (J.B.); wanglei0622@cau.edu.cn (L.W.)

**Keywords:** anaerobic fermentation, feed stuff, transgenic engineered lactic acid bacteria, fermentation quality

## Abstract

**Simple Summary:**

The silage fermentation of *Stylosanthes* is one of the most effective solutions to solve the shortage of feed due to the inability of *Stylosanthes* to grow in winter. In our previous study, it was found that the effect of direct silage fermentation was poor due to factors such as the high buffer energy value and high fiber content. In this study, we used the transgenic engineered lactic acid bacteria independently developed by our team as additives to explore the effects of cellulase-producing engineered lactic acid bacteria on the fermentation quality and in vitro digestibility of *Stylosanthes* silage under different raw material moisture contents. The results are discussed in terms of chemical composition. We found that lactic acid bacteria can produce a large amount of cellulase in the process of *Stylosanthes* silage fermentation, significantly reduce the fiber content in *Stylosanthes*, and improve the quality and in vitro digestibility of *Stylosanthes* silage. Our research results provide a deeper understanding of the influence of moisture content and lactic acid bacteria additives on *Stylosanthes* silage, and provide technical support and a theoretical basis for guiding production practice and further in-depth research, development and utilization of more warm-season forage silage.

**Abstract:**

The present study aims to estimate the dynamic effects of moisture levels and inoculants on the fermentation quality and in vitro degradability of *Stylosanthes* silage. In this experiment, *Stylosanthes* was ensiled with (1) no additive (control), (2) *Lactobacillus plantarum* (LP), (3) *Lactobacillus plantarum* carrying heterologous genes encoding multifunctional glycoside hydrolases (xg), or (4) LP + xg and was wilted until different moisture levels (60% and 72%) were attained. The ensiled bags were unpacked after different storage periods to determine the chemical composition and fermentation quality of the *Stylosanthes* silage. Moreover, the in vitro degradability was also determined 45 days after the ensiling process. The results show that the silage prepared with freshly mowed *Stylosanthes* also had a lower pH and NH_3_- N content. Adding transgenic engineered lactic acid bacteria xg not only decreased the NDF and ADF content of the silage, but also improved the in vitro digestibility significantly. We concluded that the addition of xg to *Stylosanthes* silage can improve its quality and increase in vitro digestibility and gas production. The results provide technical support and a theoretical basis for the utilization of warm-season forage silage.

## 1. Introduction

The shortage of high-quality forage grass and protein feed is an important factor restricting the development of animal husbandry. Alfalfa, a high-quality protein forage resource, has been planted and utilized on a large scale worldwide [1]. *Stylosanthes guianensis*, a common flowering legume that is native to South America, grows mainly in tropical and subtropical regions. In the subtropics, it is considered an important feed source for ruminants, with high yields, high nutrient levels, and wide adaptability [2]. However, the surface of the *Stylosanthes* stalk is rough and hairy, and its palatability is poor. In addition, *Stylosanthes* is produced seasonally. If it cannot be harvested and used in time during its peak growth period in the summer, *Stylosanthes* will continue to grow until it ages, and its feed value will decrease, resulting in an insufficient supply of livestock feed in the winter and spring seasons [3]; therefore, storing it as silage would be highly beneficial. Silage quality is affected by many factors, and legumes such as *Stylosanthes* are difficult to ensile directly without additives because of their high buffering capacity, low water-soluble carbohydrate content, and high dry matter content [4].

During the silage process, the lactic acid bacteria attached to the surface of raw materials are fermented under anaerobic conditions to convert water-soluble carbohydrates (WSCs) into organic acids, mainly lactic acid. This rapidly acidifies the silage environment and inhibits the growth of harmful microorganisms; thus, the feed can be preserved for a long time [5]. *Stylosanthes* has characteristics such as a high buffering energy, low sugar content, and high moisture content at harvest, which are similar to those of alfalfa; hence, it is more difficult to prepare silage directly [6,7]. The use of additives is important for preparing high-quality silage. The use of lactic acid bacteria is in accordance with the requirements of sustainable development and is attractive to consumers because of their advantages, which include their non-toxicity, harmlessness, cost-effectiveness, and ease of use; they are widely used in production practices. The added lactic acid bacteria multiply rapidly in the early stage of the silage process, produce lactic acid, improve fermentation quality, and reduce nutrient loss [8].

In recent years, with the development of molecular biology techniques and the continuous in-depth research on the mechanism of action of cellulase, an increasing number of reports have emerged regarding the use of food-safe lactic acid bacteria for the expression of exogenous cellulase genes. Hu et al. [9] successfully introduced the xylanase gene (*xynR8*) from rumen fungi into *Lactobacillus brevis* and detected that the enzyme activity was 0.412 U/mL, and that it could use xylan to produce lactic acid. Our laboratory used molecular biology techniques to successfully introduce the cellulase-expressing gene *xg* into *Lactobacillus plantarum* (LP) via genetic engineering technology, and successfully constructed cellulase-producing engineered lactic acid bacteria (LP-p11-*celA*-*xg, xg*); their application on alfalfa silage resulted in positive results. These bacteria not only reduced the content of neutral detergent fiber, acid detergent fiber, and acid detergent lignin in alfalfa silage to a significant extent, but also effectively inhibited protein degradation, improved in vitro digestibility and gas production, and significantly improved the quality of alfalfa silage [10]. Fiber digestibility is the main factor limiting the efficient use of silage by ruminants. Many early laboratory silo-scale studies have been conducted to assess the effect of LP on the digestibility of forages, in a manner similar to that observed for protein fractions, and the results have often been inconsistent.

To the best of our knowledge, no research has investigated the effect of transgenic engineered lactic acid bacteria on *Stylosanthes* silage. Based on the previous research results on alfalfa silage, we speculate that the cellulase-producing engineered lactic acid bacteria can release cellulase during the fermentation process to reduce the fiber content in *Stylosanthes*, which is also a leguminous forage, and may further improve the quality and digestive performance of *Stylosanthes* silage. Therefore, the purpose of this study is to explore the effect of the addition of lactic acid bacteria on the quality of *Stylosanthes* silage, in vitro digestion, and gas production. Our findings provide a basis for the development of production practices and the subsequent utilization of greater quantities of warm-season pasture silage.

## 2. Materials and Methods

### 2.1. Silage Materials and Ensiling

The test site for this experiment is located at the Tropical Forage Experiment Base of the Tropical Crop Variety Resources Research Institute, Chinese Academy of Tropical Agricultural Sciences, Danzhou City, Hainan Province (19°30′ N, 109°30′ E, altitude 149 m). The test site exhibits a tropical monsoon climate, i.e., dry and wet conditions. The soil of the experimental base is brick red loam developed from granite, and the soil texture is poor. The raw material used was obtained during the third vegetative period of *Stylosanthes* mowed on 3 July 2019; the variety number was TF380.

The *Stylosanthes* raw materials were cut into 1–2 cm sections with an artificial guillotine, mixed thoroughly with additives, and stored in a transparent polyethylene silage bag (250 × 300 mm) that were sealed with a vacuum packaging machine. The bags were stored at room temperature (25–30 °C) away from light; each bag weighed about 300 g. After 1, 3, 7, 15, and 45 days of storage, the bags were opened for sampling analysis.

Part of the *Stylosanthes* material was freshly cut and silaged directly (the moisture content was about 72%); the other part was silaged when the moisture content was about 60% after being properly dried. In this experiment, *Stylosanthes* was ensiled with (1) no additive (control), (2) LP, (3) LP carrying heterologous genes encoding multifunctional glycoside hydrolases (xg), or (4) LP + xg and was wilted until different moisture levels (60% and 72%). The additives were provided by the Forage Processing and Utilization Laboratory of China Agricultural University, and the added amount was 1 × 10^6^ cfu/g fresh grass; the same amount of distilled water was added to the control group. Each treatment process was performed three times.

### 2.2. Chemical Analysis

The bag was opened, the *Stylosanthes* silage was mixed well, and two subsamples weighing 20 g were obtained from each bag and homogenized with 180 mL of distilled water for 2 min in a blender jar. The extracts were filtered through four layers of cheesecloth and filter paper. The filtrate was used to determine the pH, as well as the ammonia nitrogen (NH_3_-N) and organic acid concentrations. The pH was measured using a pH meter (PHS-3C, INESA Scientific Instrument Co., Ltd., Shanghai, China), and the NH_3_-N content was determined using the sodium hypochlorite and phenol method [11]. The determination of organic acid content was performed using high-performance liquid chromatography (HPLC) [12]. Two subsamples with a weight of 200 g were obtained from each bale and dried in a forced-draft oven at 65 °C for 48 h to determine dry matter (DM) content. The dried samples were ground and passed through a 1.0-mm screen for chemical analysis. Water-soluble carbohydrate (WSC) levels were determined via the anthrone method [13]. Crude protein (CP) levels were determined using method 984.13 of the Association of Official Analytical Chemists [14]. The amylase-treated neutral detergent fiber (aNDF) and acid detergent fiber (ADF) levels were analyzed according to the method described by Van Soest et al. [15].

### 2.3. In Vitro Incubation and Degradability Measurement

In vitro fermentation was carried out in Ankom RFS bottles using the pressure transducer technique (Ankom Technologies, Macedon, NY, USA), as described by Yuan et al. [16]. Cumulative gas production data were fitted to a gas production model modified from the following Gompertz growth equation [17]. In vitro DMD and neutral detergent fiber digestibility (NDFD) were determined with an Ankom DaisyⅡincubator (Ankom Technologies, Macedon, NY, USA). Approximately 0.5 g of ground samples was added into the same fluid–buffer mixtures and incubated in the presence of CO_2_ for 48 h. The reduced weight of *Stylosanthes* silage samples was used to calculate the in vitro DMD. The aNDF content of the residue after incubation was also determined and used to calculate the in vitro NDFD.

### 2.4. Statistical Analyses

The data regarding the fermentation characteristics, chemical composition, and in vitro degradability were analyzed using SPSS version 19.0 for Windows (SPSS Inc., Chicago, IL, USA). Duncan’s multiple range method was used to identify differences among the means of treatments when the level of interaction was significant. Means were considered significantly different at *p* < 0.05. Gas production-related kinetic parameters such as V (∞) and k were also estimated with an iterative least square method using a non-linear regression SPSS procedure.

## 3. Results

### 3.1. Fermentation Quality of Stylosanthes Silage

The fermentation quality-related dynamics of high-moisture content *Stylosanthes* silage are shown in Table 1. Throughout the fermentation process, the pH value of each treatment group showed a significant decreasing trend (*p* < 0.05). On the 45th day, the pH value of the LP + xg treatment group was the lowest.

During the entire fermentation process, the lactic acid content of each treatment group showed a significant increasing trend (*p* < 0.05). Fifteen days before fermentation, the lactic acid content of each treatment group increased rapidly, stabilized after the 15th day of fermentation, and peaked on the 45th day of fermentation. During the early stage of silage fermentation, the lactic acid content of each treatment group was similar.

The acetic acid content of each treatment group increased significantly during the entire silage fermentation process (*p* < 0.05); it increased rapidly in each treatment group 7 days before fermentation and stabilized after 7 days of fermentation. On the 45th day of fermentation, the acetic acid levels of the treatment groups xg and LP + xg were significantly lower than those of the control group and LP treatment group (*p* < 0.05), while the acetic acid content of the xg treatment group was the lowest.

With the increase in silage fermentation time, the propionic acid content in each treatment group showed a significant increase (*p* < 0.05); at the same fermentation time, there was no significant difference in the propionic acid content between treatment groups (*p* > 0.05). No butyric acid was detected during the entire process of silage fermentation.

Within days 1–45 of fermentation, the ammonia nitrogen content for each treatment group showed a significant increasing trend (*p* < 0.05) with an increase in the fermentation time. At the beginning of the fermentation process, there was no significant difference in the ammonia nitrogen content between treatment groups (*p* > 0.05). Subsequently, the ammonia nitrogen content in the control group became significantly higher than that in the three treatment groups (*p* < 0.05).

The fermentation quality dynamics of *Stylosanthes* silage with low moisture contents are shown in Table 2. During the first 1–7 days of fermentation, the pH value of each treatment group decreased gradually. On the 45th day, the pH value of the control was significantly higher than that of the other three treatment groups (*p* < 0.05), and the pH value of the LP treatment group was significantly higher than that of xg and LP + xg (*p* < 0.05). There was no significant difference in the pH value between the LP, LP + xg, and control groups, as compared to that of the xg (*p* > 0.05) group; the values were 5.06, 4.95, 4.64, and 4.58, respectively.

During the first 15 days of fermentation, with an increase in the fermentation time, the lactic acid content of each treatment group increased significantly (*p* < 0.05); it tended to be stable between the 15th to 45th day of fermentation, and there was hardly any change in the lactic acid content.

From the 1st to the 45th day of the fermentation process, the acetic acid content in the control group did not change significantly (*p* > 0.05) with an increase in the silage fermentation time. On the 45th day of fermentation, the acetic acid content in the control group was significantly higher than that in the three additive treatment groups (*p* < 0.05), and there was no significant difference between the three additive treatment groups (*p* > 0.05). The acetic acid content in the LP + xg treatment group was the lowest.

From the 1st to the 45th day of fermentation, with an increase in the silage fermentation time, the propionic acid content in each treatment group showed a significant increase (*p* < 0.05). On the 45th day of fermentation, the propionic acid content of the control group was the highest, and that of the LP + xg treatment group was the lowest. No butyric acid was detected during the entire process of silage fermentation.

Within days 1–45 of fermentation, the ammonia nitrogen content in each treatment group showed a significant increasing trend (*p* < 0.05) with an increase in the fermentation time. At the initial fermentation stage, there was no significant difference in the ammonia nitrogen content between the treatment groups (*p* > 0.05). Subsequently, the ammonia nitrogen content in the control group became significantly higher than that in the three additive treatment groups (*p* < 0.05). There was no significant difference among the treatment groups (*p* > 0.05).

The effects of different factors and the interactions between them on the fermentation quality of *Stylosanthes* silage are shown in Table 3. Moisture content had a very significant effect on the pH value and the content of NH_3_-N, lactic acid, acetic acid, and propionic acid (*p* < 0.01). Treatment had a very significant effect on the pH value and the NH_3_-N, lactic acid, and acetic acid content (*p* < 0.01), but had no significant effect on the propionic acid content (*p* > 0.05). Ensiling time had a very significant effect on the pH value and the NH_3_-N, lactic acid, acetic acid, and propionic acid content (*p* < 0.01). The interaction between moisture content and treatment as well as the interaction between treatment and ensiling time had a very significant effect on the pH value and the NH_3_-N, lactic acid, and acetic acid (*p* < 0.01) content, but had no significant effect on the propionic acid content (*p* > 0.05). The interaction between the moisture content and ensiling time had a very significant effect on the pH value and the NH_3_-N, lactic acid, acetic acid, and propionic acid (*p* < 0.01) content. The interactions among these three processing factors had a very significant effect on the pH value and the NH_3_-N, lactic acid, and acetic acid (*p* < 0.01) content, but had no significant effect on the propionic acid (*p* > 0.05) content.

### 3.2. Chemical Compositions of Stylosanthes Silages

The chemical composition of *Stylosanthes* before the silage process is shown in Table 4.

Table 5 shows the effects of additives on nutrient quality dynamics in high-moisture content *Stylosanthes* silage. The DM content in *Stylosanthes* silage exhibited a relatively stable changing trend during the entire fermentation process. At the initial stage of silage fermentation, there was no significant difference in the DM content of *Stylosanthes* silage between treatment groups (*p* > 0.05).

The CP content in *Stylosanthes* silage exhibited a relatively stable changing trend during the entire fermentation process, and no significant difference in the extent of change was observed (*p* > 0.05). At the same silage fermentation time, there was no significant difference in the CP content between the treatment groups (*p* > 0.05).

During days 1–45 of silage fermentation, the WSC content in each treatment group showed a significant decreasing trend (*p* < 0.05). In the first 15 days of silage fermentation, with an increase in the silage fermentation time, the WSC content in each treatment group decreased rapidly and began to stabilize after 15 days of fermentation. During the entire fermentation process, the WSC content of the xg and LP + xg treatment groups decreased to a lesser extent. At a later stage of fermentation, the WSC content of the xg treatment group was significantly higher than that of the LP + xg (*p* < 0.05) and LP groups (*p* < 0.05). The WSC content in the treatment groups was significantly higher than that in the control group (*p* < 0.05)

During days 1–45 of silage fermentation, with an increase in the fermentation time, the NDF content in each treatment group showed a significant decreasing trend (*p* < 0.05). On the 45th day of fermentation, the NDF levels in each treatment group were the lowest. Of these, the NDF content of the xg treatment group was the lowest and was significantly lower than that of the three other treatment groups (*p* < 0.05).

During the silage fermentation process, for 1–15 days, the ADF content of each treatment group showed a significant decreasing trend (*p* < 0.05) with an increase in the fermentation time. During the entire fermentation process, the ADF content of the xg and LP + xg treatment groups was lower than that of the control group and the LP treatment group. By the 45th day of silage fermentation, the ADF content in the xg treatment group was significantly lower than that in the three other treatment groups (*p* < 0.05).

Table 6 shows the effects of the additives on the nutrient quality dynamics of *Stylosanthes* silage with low moisture contents. The DM content of *Stylosanthes* silage in each treatment group gradually increased with an increase in the silage fermentation time. In the early stage of silage fermentation, the difference in the DM content between the treatment groups was not significant (*p* > 0.05). At day 45 of silage fermentation, the DM content in the LP, xg, and LP + xg additive treatment groups was significantly lower than that of the control group (*p* < 0.05).

The CP content in *Stylosanthes* silage exhibited a relatively stable changing trend during the entire fermentation process, and no significant differences were observed in the changes (*p* > 0.05). At the same silage fermentation time, there was no significant difference in the CP content among the treatment groups (*p* > 0.05).

During the process of silage fermentation for 1–15 days, it was observed that the WSC content of each treatment group decreased rapidly with an increase in the silage fermentation time, and began to stabilize after 15 days of fermentation. By the 45th day of fermentation, the WSC content of the xg treatment group was significantly higher than that of the other three groups (*p* < 0.05).

During the process of silage fermentation for 1–45 days, it was observed that the change in the NDF content in each treatment group showed a significant decreasing trend (*p* < 0.05) with an increase in the fermentation time. By the 45th day of fermentation, the NDF content was the lowest, and was significantly lower than that of the control group and the LP treatment group.

With an increase in the silage fermentation time, the trend of changes observed for the ADF content in the LP treatment group and control group was relatively stable, while those in the xg and LP + xg treatment groups decreased significantly (*p* < 0.05). By the 45th day of fermentation, the ADF content in the LP + xg treatment group was the lowest, and was significantly lower than that in the LP treatment group and control group (*p* < 0.05).

The effects of different factors and the interactions between them on the chemical compositions of *Stylosanthes* silage are shown in Table 7. Moisture content had a very significant effect on the DM, CP, WSC, NDF, and ADF (*p* < 0.01) content. Treatment had a very significant effect on the DM, WSC, NDF, and ADF (*p* < 0.01) content, but had no significant effect on the CP content (*p* > 0.05). Ensiling time had a very significant effect on the DM, WSC, NDF, and ADF (*p* < 0.01) content, and this had a significant effect on the CP content (*p* < 0.05). The interaction between moisture content and treatment as well as the interaction between moisture content and ensiling time had a very significant effect on the DM, CP, WSC, NDF, and ADF (*p* < 0.01) content. The interaction between treatment and ensiling time had a very significant effect on the CP, WSC, NDF, and ADF (*p* < 0.01) content, and a significant effect on the DM content (*p* < 0.05). The interactions among these three processing factors had a very significant effect on the DM, CP, WSC, NDF, and ADF (*p* < 0.01) content.

### 3.3. Effects of Moisture Content and Additives on the In Vitro Digestibility of Stylosanthes Silage

It can be seen from Table 8 that the effect of moisture content on the in vitro DM digestibility of *Stylosanthes* silage for 48 h was not significant (*p* > 0.05). The effect of additives on the in vitro DM digestibility of *Stylosanthes* silage for 48 h was extremely significant (*p* < 0.01). The interaction between the moisture content and additives resulted in no significant differences in the 48 h in vitro DM digestibility of *Stylosanthes* silage (*p* > 0.05).

Among the four treatment groups, the 48 h in vitro DM digestibility of xg *Stylosanthes* silage was the highest, i.e., 67.59% and 68.57%; these values were significantly higher than those of the other three treatment groups (*p* < 0.05). The second-highest value was observed for the LP + xg treatment group, and was significantly higher than that observed with LP (*p* < 0.05). The values for the three additive treatment groups were significantly higher than that for the control group (*p* < 0.05).

It can be seen from Table 9 that the effect of moisture content on the 48 h in vitro digestibility of neutral detergent fiber of *Stylosanthes* silage was not significant (*p* > 0.05). The effect of additives on the 48 h in vitro neutral detergent fiber digestibility of *Stylosanthes* silage was extremely significant (*p* < 0.01). The interaction between the moisture content and additives resulted in no significant differences in the 48 h in vitro neutral detergent fiber digestibility of *Stylosanthes* silage (*p* > 0.05).

Among the four treatment groups, the 48 h in vitro neutral detergent fiber digestibility of xg *Stylosanthes* silage was the highest, i.e., 38.31% and 38.93%. These values were significantly higher than those observed for the three other treatment groups (*p* < 0.05). The second-highest values were observed for the LP + xg treatment group, and were significantly higher than that observed for LP (*p* < 0.05). The 48 h in vitro neutral detergent fiber digestibility of the control group was the lowest, i.e., 27.01% and 27.49%.

### 3.4. Effects of Moisture Content and Additives on the In Vitro Gas Production by Stylosanthes Silage

It can be seen from Table 10 that additive treatment can significantly increase the in vitro gas production by *Stylosanthes* silage (*p* < 0.01). The effect of the addition of xg was the most obvious, and the addition of LP + xg resulted in the second-most notable effect. The effect of moisture content on the in vitro gas production of *Stylosanthes* silage was not significant (*p* > 0.05). The interaction of moisture content and additives resulted in no significant differences in the in vitro gas production by *Stylosanthes* silage (*p* > 0.05).

The effects of additives on in vitro gas production in the rumen of high-moisture content *Stylosanthes* silage are shown in Figure 1. In the initial stage, the additives did not have a significant effect on in vitro gas production by *Stylosanthes* silage (*p* > 0.05). At 4 h and 6 h, the in vitro gas production by the xg treatment group was significantly higher than that of the three other groups (*p* < 0.05). There was no significant difference in the in vitro gas production between the LP treatment group and the control group (*p* > 0.05). By 36 h, the in vitro gas production had gradually stabilized in *Stylosanthes* silage. The in vitro gas production in each group was the highest at 48 h. The xg treatment group exhibited the highest in vitro gas production, followed by the LP + xg treatment group and the LP treatment group. The in vitro gas production in the three treatment groups was significantly higher than that of the control group (*p* <0.05).

The effects of additives on in vitro gas production in the rumen of low-moisture content *Stylosanthes* silage are shown in Figure 2. At 2 h, the additives did not have a significant effect on in vitro gas production by *Stylosanthes* silage (*p* > 0.05). From 4 h to 12 h, the in vitro gas production in the xg and LP + xg treatment groups was significantly higher than that in the LP treatment group and control group (*p* < 0.05). There was no significant difference in in vitro gas production between the LP treatment group and the control group (*p* > 0.05). From 24 h to 48 h, the in vitro gas production in the xg treatment group was the highest, followed by the LP + xg treatment group and the LP treatment group. The in vitro gas production in the three treatment groups was significantly higher than that of the control group (*p* < 0.05). From 36 h to 48 h, there was hardly any difference in the in vitro gas production among the groups, and it gradually stabilized and reached the highest value.

## 4. Discussion

The moisture content of the *Stylosanthes* raw material is a very important factor influencing the *Stylosanthes* silage process [18,19,20]. When the moisture content of the raw material is 60–72%, it is more conducive to the fermentation of lactic acid bacteria during [21]. In this study, the moisture content of the freshly cut *Stylosanthes* was about 72%, and the moisture content of the properly aired *Stylosanthes* was about 60%. The fermentation quality of *Stylosanthes* silage with higher raw material moisture content was better than that with lower moisture content under the same additive treatment, which was manifested by the lower pH and the lower NH_3_-N content and higher lactic acid content. This result may be due to the harvesting period of the *Stylosanthes*, so when the moisture content is about 72%, the *Stylosanthes* can be directly silaged. This also took place in the study by Zou et al. [22]. Research has shown that wilting can decrease the availability of inorganic ions, and form a buffer system with weak organic acids in the silage [7]. In general, good silage depends on reasonable moisture contents of the material, and wilted silage had poor aerobic stability [18]. Reducing moisture contents increases silage pH and decreases lactic, acetic and propionic acid yield [23].

Silage raw materials with adequate moisture levels can not only improve the fermentation quality of the silage, but also reduce silage permeate production and nutrient loss. The effect of the degree of withering on the quality of ryegrass silage was analyzed; it was observed that when the moisture content was 73%, no silage permeate was produced, and better silage quality could be achieved. The use of additives can substantially enhance silage quality.

In this study, the three additive treatment groups all had a significant effect (*p* < 0.05) compared to the control group. Among these, the pH value of the xg and LP + xg treatment groups was between 4.33 and 4.64, the lactic acid content was above 4%, and the ammonia nitrogen/total nitrogen content was below 10%. This may be attributable to the fact that the addition of LP has a better effect on *Stylosanthes* silage that was similar to that observed upon the addition of LP to alfalfa silage, to improve the fermentation quality, as described by Contreras [24]. After the addition of xg, the cellulase expressed by xg and lactic acid bacteria together on the *Stylosanthes* raw materials produced a large number of organic acids, mainly lactic acid. These organic acids reduced the pH value during the silage process, thereby effectively inhibiting the growth of undesirable microorganisms, such as yeast and clostridia, in order to improve the fermentation quality of silage. The results of this study are similar to the conclusions derived by Zahiroddini et al. [25] after the addition of lactic acid bacteria and cellulase to barley silage. The content of ammonia nitrogen/total nitrogen reflects the degree of decomposition of protein and amino acids in the silage—an increase in its content will significantly reduce the feed intake of livestock, which has a negative impact on milk production and milk quality [26]. In this study, under the same treatment, the moisture content of silage raw materials greatly affected the nutritional quality of *Stylosanthes* silage. If the moisture content of the raw materials decreased, the WSC content in the silage decreased, and the NDF and ADF levels increased. The WSC and DM levels decreased with an increase in the silage fermentation time. This may be attributable to the fact that the fermentation and respiration of anaerobic microorganisms reduce the DM and WSC content in a sealed environment [27,28]. The CP content of the control group was similar to that observed before the silage process, and the difference was small, but the ammonia nitrogen content was significantly increased, possibly because the true protein in the control group was degraded into non-protein nitrogen.

The addition of xg and LP + xg can significantly reduce the NDF and ADF content in *Stylosanthes* silage compared to that in the control and LP treatment groups. This significantly increased the WSC content in the *Stylosanthes* silage (*p* < 0.05), probably because the production of xg can degrade the cell wall of *Stylosanthes* and reduce its fiber content, thus producing WSC. This result is similar to the results of a study by Lynch et al., in which cellulase was added to forage [29]. The conclusion that an increase in the WSC content is consistent with a decrease in the NDF and ADF content is also consistent with the conclusion that the addition of cellulase to legume silage by Foster et al. [30] increased the WSC content.

The DM digestibility of silage can represent the ability of the feed to be utilized and transformed by microorganisms. In vitro DM digestibility can reflect the degree of degradation of the feed by rumen microorganisms in the fermentation system. The substrate for in vitro fermentation gas production is mainly WCS, which can reflect the utilization of substrates by rumen microorganisms and reflect the nutritional value of feed [31]. DM digestibility is also an important factor affecting DM intake. The higher the DM digestibility, the greater the DM intake of ruminants. In this study, the treatment group with a higher content of neutral detergent fiber and acid detergent fiber had lower DM digestibility. Thus, we concluded that in vitro DM digestibility is significantly negatively correlated with the content of neutral detergent fiber and acid detergent fiber; our conclusions are the same as those of Broderick et al. [11]. If other conditions remain unchanged, the digestibility of neutral detergent fiber is an important indicator of the nutritional value of roughage. Improving the digestibility of neutral detergent fiber can significantly increase the DM intake of ruminants. Zhu [32] found that adding lactic acid bacteria to alfalfa silage can significantly improve the in vitro digestibility of alfalfa silage. In this research, lactic acid bacteria additives had a significant effect on the in vitro DM digestibility, in vitro neutral detergent fiber digestibility, and in vitro gas production of *Stylosanthes* silage. This conclusion is similar to that derived in a study by Bureenok et al. [33].

Research has shown that differences in the chemical composition and the contents of NDF and ADF alter IVDMD and IVNDFD [34]. In vitro gas production measurements are an effective way to detect silage quality. Thus, many researchers have evaluated the digestibility of feed by this technology [35]. The amount of in vitro gas production was closely related to the rate of degradation of carbohydrates contained in the substrate. The larger the amount of gas produced, the better the degree of substrate fermentation and the higher the degree of rumen digestion. Previous research showed that in vitro gas production was positively correlated with IVDMD [36,37], which was similar to the results of our study. Studies have shown that in vitro fermentation gas yield is negatively correlated with the NDF content in fermentation substrates. Theodorou et al. [38] established a simple and accurate measurement—barometric pressure conversion technology—to evaluate ruminant feed in vitro. In this study, in vitro gas production by *Stylosanthes* silage increased with time and began to stabilize after 36 h; these results are consistent with the results of previous studies. Nsahlai et al. [39] found that in legume pastures, in vitro gas production by silage was significantly negatively correlated with the neutral detergent fiber content. In this study, the treatment group with a lower NDF content exhibited a higher level of in vitro gas production, which is consistent with this conclusion. The addition of xg can significantly increase the in vitro digestibility and in vitro gas production of *Stylosanthes* silage. This may be attributable to the fact that the cellulase and cellobiose expressed by xg can degrade the cell wall of forage grass, which could result in the release of more cell content, to provide energy for the growth of rumen microorganisms. On the other hand, the released cellulase destroyed the structural barrier of forage grass, improved the adhesion of rumen microorganisms to the substrate, and increased the in vitro gas production of silage. This conclusion is similar to that derived in a study by Lazark et al. [40], in which cellulase was added to alfalfa silage.

## 5. Conclusions

The silage prepared with freshly mowed *Stylosanthes* also had better quality in the present study. Therefore, when preparing *Stylosanthes* silage, the raw materials did not need to be aired. Adding transgenic engineered lactic acid bacteria xg not only improved the quality of *Stylosanthes* silage, but also improved the in vitro digestibility significantly. This work may provide valuable insights into the processing and modulation of *Stylosanthes* silage, which can be processed into high-quality silage with a higher fiber content and greatly improve the utilization efficiency of *Stylosanthes*. In addition, it also provides a theoretical basis for the development and utilization of other warm-season grass silages with high fiber content. In future research, the community composition and metabolites of *Stylosanthes* silage can also be determined through microbial sequencing and metabonomics technology, and the fermentation of *Stylosanthes* silage can be more comprehensively explored.

## Figures and Tables

**Figure 1 animals-12-01555-f001:**
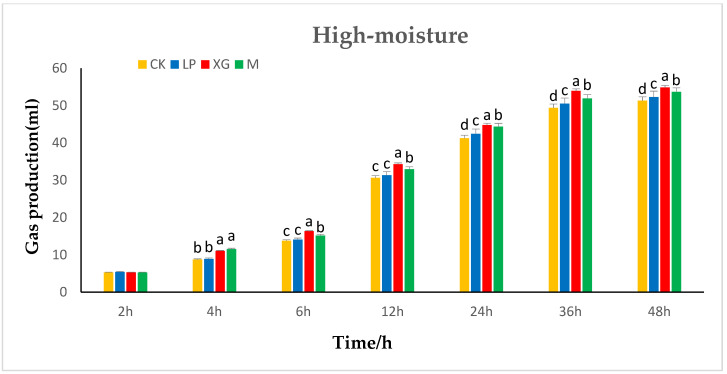
Effects of additives on the in vitro gas production in rumen of high-moisture content *Stylosanthes* silage.

**Figure 2 animals-12-01555-f002:**
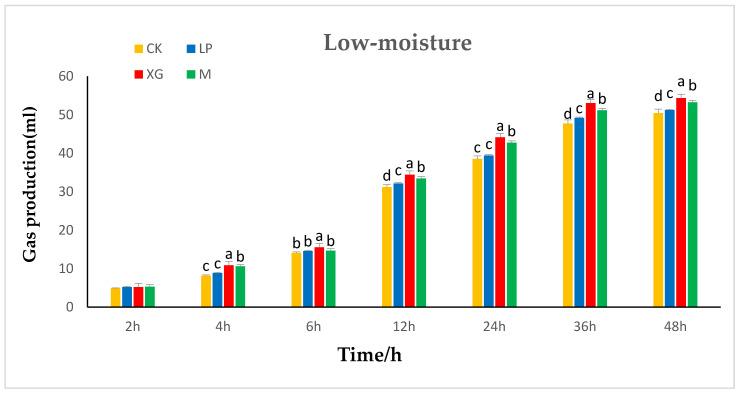
Effects of additives on the in vitro gas production in rumen of low-moisture content *Stylosanthes* silage.

**Table 1 animals-12-01555-t001:** Changes in fermentative characteristics during the ensiling of high-moisture content *Stylosanthes* silages.

Item ^1^	Treatment ^2^	Days of Ensiling					SEM	*p*-Value ^3^		
		1	3	7	15	45		D	T	D ∗ T
Fermentative Characteristics										
pH	Control	5.68 ^aA^	5.5 ^aB^	5.49 ^aB^	4.95 ^aC^	4.94 ^aC^	0.76	<0.001	<0.001	<0.001
	LP	5.6 ^bA^	5.41 ^bB^	5.35 ^bC^	4.66 ^bD^	4.59 ^bE^				
	xg	5.51 ^cA^	5.01 ^cB^	4.99 ^cB^	4.51 ^cC^	4.47 ^cC^				
	LP + xg	5.45 ^dA^	4.99 ^cB^	4.94 ^cB^	4.39 ^dC^	4.33 ^dC^				
Lactic acid (g kg^−1^ DM)	Control	6.13 ^bE^	12.61 ^cD^	19.69 ^cC^	33.14 ^dB^	34.32 ^dA^	0.53	<0.001	<0.001	<0.001
	LP	7.76 ^aE^	13.43 ^bD^	23.44 ^bC^	36.63 ^cB^	38.93 ^cA^				
	xg	8.33 ^aE^	14.91 ^aD^	25.63 ^aC^	43.92 ^bB^	45.74 ^bA^				
	LP + xg	8.16 ^aE^	15.33 ^aD^	26.41 ^aC^	45.11 ^aB^	48.26 ^aA^				
Acetic acid (g kg^−1^ DM)	Control	19.32 ^a^	19.62 ^a^	21.36 ^a^	20.12 ^a^	19.43 ^a^	0.39	<0.001	<0.001	<0.001
	LP	17.93 ^bBC^	22.08 ^abA^	19.81 ^abB^	18.24 ^bBC^	16.88 ^bC^				
	xg	17.34 ^bC^	19.73 ^bA^	18.58 ^bAB^	17.83 ^bBC^	16.37 ^bC^				
	LP + xg	17.41 ^bAB^	19.14 ^bA^	18.13 ^bA^	18.77 ^abA^	15.42 ^bB^				
Propionic acid (g kg^−1^ DM)	Control	14.23 ^C^	13.93 ^C^	14.41 ^C^	16.18 ^B^	19.93 ^A^	0.62	<0.001	0.315	0.588
	LP	14.27 ^C^	15.08 ^BC^	14.33 ^C^	16.16 ^B^	18.84 ^A^				
	xg	14.41 ^C^	14.54 ^C^	13.84 ^C^	16.33 ^B^	18.31 ^A^				
	LP + xg	14.38 ^C^	14.83 ^C^	14.93 ^C^	16.59 ^B^	17.91 ^A^				
NH_3_-N (g kg^−1^ TN)	Control	5.33 ^E^	21.73 ^aD^	52.88 ^aC^	84.73 ^aB^	154.49 ^aA^	0.47	<0.001	<0.001	<0.001
	LP	5.51 ^E^	12.11 ^bD^	26.36 ^bC^	37.49 ^bB^	75.32 ^cA^				
	xg	5.52 ^E^	12.22 ^bD^	24.74 ^bC^	39.44 ^bB^	74.61 ^cA^				
	LP + xg	5.59 ^E^	13.17 ^bD^	24.61 ^bC^	38.73 ^bB^	80.24 ^bA^				

Means within the same row (A–E) or within the same column (a–d) with different superscripts differ significantly from each other (*p* < 0.05). SEM, standard error of the mean. ^1^ NH_3_-N, ammonia nitrogen; TN, total nitrogen; DM, dry matter. ^2^ LP, *Lactobacillus plantarum* inoculant; xg, *LP-p118200-celA-xg* inoclulant. ^3^ D, effect of ensiling days; T, effect of treatment; D ∗ T, interaction between ensiling days and treatment.

**Table 2 animals-12-01555-t002:** Changes in fermentative characteristics during the ensiling of low-moisture content *Stylosanthes* silages.

Item ^1^	Treatment ^2^	Days of Ensiling					SEM	*p*-Value ^3^		
		1	3	7	15	45		D	T	D ∗ T
Fermentative Characteristics										
pH	Control	5.52 ^bA^	5.49 ^aB^	5.47 ^aB^	5.05 ^aC^	5.06 ^aC^	0.39	<0.001	<0.001	<0.001
	LP	5.51 ^bA^	5.48 ^AB^	5.45 ^aB^	4.92 ^bC^	4.95 ^bC^				
	xg	5.51 ^bA^	5.38 ^bB^	5.31 ^bC^	4.7 ^cD^	4.64 ^cE^				
	LP + xg	5.49 ^aA^	5.36 ^bB^	5.28 ^bC^	4.68 ^cD^	4.58 ^cE^				
Lactic acid (g kg^−1^ DM)	Control	6.13 ^bE^	12.63 ^cD^	19.74 ^cC^	33.06 ^dB^	34.34 ^dA^	0.91	<0.001	<0.001	<0.001
	LP	7.78 ^aE^	13.44 ^bD^	23.38 ^bC^	36.62 ^cB^	38.93 ^cA^				
	xg	8.26 ^aE^	14.91 ^aD^	25.63 ^aC^	43.92 ^bB^	45.72 ^bA^				
	LP + xg	8.13 ^aE^	15.27 ^aD^	26.37 ^aC^	45.13 ^aB^	48.31 ^aA^				
Acetic acid (g kg^−1^ DM)	Control	19.34 ^a^	19.63 ^a^	21.38 ^a^	20.14 ^a^	19.42 ^a^	1.31	<0.001	<0.001	<0.001
	LP	17.91 ^bBC^	22.14 ^abA^	19.82 ^abB^	18.21 ^bBC^	16.91 ^bC^				
	xg	17.38 ^bC^	19.73 ^bA^	18.63 ^bAB^	17.79 ^bBC^	16.38 ^bC^				
	LP + xg	17.26 b^AB^	19.14 ^bA^	18.11 ^bA^	18.82 ^abA^	15.41 ^bB^				
Propionic acid (g kg^−1^ DM)	Control	14.21 ^C^	13.92 ^C^	14.42 ^C^	16.27 ^B^	19.87 ^A^	0.81	<0.001	0.127	0.226
	LP	14.32 ^C^	15.12 ^BC^	14.31 ^C^	16.26 ^B^	18.76 ^A^				
	xg	14.37 ^C^	14.53 ^C^	13.77 ^C^	16.33 ^B^	18.32 ^A^				
	LP + xg	14.43 ^C^	14.83 ^C^	14.93 ^C^	16.63 ^B^	17.86 ^A^				
NH_3_-N (g kg^−1^ TN)	Control	5.31 ^E^	21.71 ^aD^	52.92 ^aC^	84.71 ^aB^	154.53 ^aA^	1.22	<0.001	<0.001	<0.001
	LP	5.53 ^E^	12.14 ^bD^	25.81 ^bC^	37.52 ^bB^	75.31 ^cA^				
	xg	5.51 ^E^	12.42 ^bD^	24.72 ^bC^	39.41 ^bB^	74.62 ^cA^				
	LP + xg	5.60 ^E^	13.24 ^bD^	24.62 ^bC^	38.73 ^bB^	80.24 ^bA^				

Means within the same row (A–E) or within the same column (a–d) with different superscripts differ significantly from each other (*p* < 0.05). SEM, standard error of the mean. ^1^ NH_3_-N, ammonia nitrogen; TN, total nitrogen; DM, dry matter. ^2^ LP, *Lactobacillus plantarum* inoculant; xg, *LP-p118200-celA-xg* inoclulant. ^3^ D, effect of ensiling days; T, effect of treatment; D∗T, interaction between ensiling days and treatment.

**Table 3 animals-12-01555-t003:** Effect of different study factors and their interaction on the fermentation quality of *Stylosanthes* silage.

	Factor ^1^	pH	NH_3_-N (g kg^−1^ TN)	Lactic Acid (g kg^−1^ DM)	Acetic Acid (g kg^−1^ DM)	Propionic Acid (g kg^−1^ DM)
*p*-value	M	<0.001	<0.001	<0.001	<0.001	<0.001
	T	<0.001	<0.001	<0.001	<0.001	0.315
	D	<0.001	<0.001	<0.001	<0.001	<0.001
	M ∗ T	<0.001	0.006	<0.001	<0.001	0.197
	M ∗ D	<0.001	<0.001	<0.001	<0.001	<0.001
	T ∗ D	<0.001	<0.001	<0.001	<0.001	0.588
	M ∗ T ∗ D	<0.001	<0.001	<0.001	0.001	0.753

^1^ M, effect of moisture content; T, effect of treatment; D, effect of ensiling days; D ∗ T, interaction between ensiling days and treatment; M ∗ T, interaction between moisture content and treatment. M ∗ D, interaction between moisture content and ensiling days. T ∗ D, interaction between treatment and ensiling days; M ∗ T ∗ D, interaction among moisture content, treatment and ensiling days.

**Table 4 animals-12-01555-t004:** Chemical compositions of *Stylosanthes* prior to ensiling.

Item	DM	CP	WSC	NDF	ADF
	%FM	%DM			
Wilted	397.08 ± 0.13	163.34 ± 0.11	24.47 ± 0.16	585.64 ± 0.77	384.38 ± 0.03
Un-wilted	282.63 ± 0.36	162.91 ± 0.08	35.83 ± 0.25	564.73 ± 0.43	366.86 ± 0.17

DM, dry matter; FM, fresh matter; CP, crude protein; WSC, water-soluble carbohydrate; NDF, neutral detergent fiber; ADF, acid detergent fiber.

**Table 5 animals-12-01555-t005:** Changes in chemical compositions during the ensiling of high–moisture *Stylosanthes* silages.

Item ^1^	Treatment ^2^	Days of Ensiling					SEM	*p*-Value ^3^		
		1	3	7	15	45		D	T	D ∗ T
Chemical Compositions										
Dry matter (g kg^−1^ FM)	Control	295.88 ^A^	290.34 ^bAB^	285.93 ^cAB^	285.82 ^cAB^	287.64 ^cB^	2.31	<0.001	<0.001	0.014
	LP	294.22	298.03 ^a^	294.83 ^ab^	295.31 ^ab^	298.43 ^ab^				
	xg	291.81 ^B^	293.28 ^bAB^	291.71 ^bB^	293.69 ^bAB^	296.11 ^bA^				
	LP + xg	300.53 ^AB^	300.49 ^aAB^	297.34 ^aB^	298.41 ^aB^	303.73 ^aA^				
Crude protein (g kg^−1^ DM)	Control	164.92	163.93	162.66	163.54	164.36	1.17	0.797	0.029	0.001
	LP	165.91 ^AB^	162.62 ^B^	167.28 ^A^	166.73 ^A^	166.41 ^A^				
	xg	164.63 ^B^	164.27 ^AB^	166.31 ^A^	166.47 ^A^	167.73 ^A^				
	LP + xg	164.11 ^AB^	162.52 ^B^	166.92 ^AB^	167.04 ^AB^	167.41 ^A^				
Water soluble carbohydrate (g kg^−1^ DM)	Control	35.66 ^A^	30.51 ^bB^	24.93 ^cC^	21.13 ^dD^	18.23 ^bE^	0.35	<0.001	<0.001	<0.001
	LP	35.14 ^A^	32.05 ^aB^	28.74 ^bC^	22.91 ^cD^	19.54 ^bE^				
	xg	35.08 ^A^	32.54 ^aB^	29.32 ^aC^	27.32 ^aD^	24.24 ^aE^				
	LP + xg	34.48 ^A^	32.08 ^aB^	28.77 ^abC^	25.21 ^bD^	23.13 ^aE^				
Neutral detergent fiber (g kg^−1^ DM)	Control	534.63 ^AB^	536.49 ^aA^	537.41 ^aA^	521.73 ^aC^	526.81 ^bBC^	3.32	<0.001	<0.001	<0.001
	LP	545.81 ^A^	536.11 ^aAB^	526.52 ^aBC^	522.34 ^aC^	537.42 ^aAB^				
	xg	547.62 ^A^	522.73 ^bB^	503.43 ^bC^	476.08 ^cD^	382.33 ^dE^				
	LP + xg	540.72 ^A^	524.88 ^bB^	512.91 ^bC^	504.21 ^bC^	402.48 ^cD^				
Acid detergent fiber (g kg^−1^ DM)	Control	369.61 ^A^	356.12 ^aB^	359.12 ^aB^	345.19 ^aC^	346.37 ^aC^	2.75	<0.001	<0.001	<0.001
	LP	369.53 ^A^	353.42 ^aB^	354.91 ^aB^	339.82 ^aC^	340.83 ^aC^				
	xg	341.81 ^A^	342.42 ^bA^	313.43 ^cB^	292.33 ^cC^	273.08 ^cD^				
	LP + xg	356.08 ^A^	343.31 ^bB^	325.42 ^bC^	303.74 ^bD^	287.82 ^bE^				

Means within the same row (A–E) or within the same column (a–d) with different superscripts differ significantly from each other (*p* < 0.05). SEM, standard error of the mean. ^1^ FM, fresh matter. ^2^ LP, *Lactobacillus plantarum* inoculant; xg, *LP-p118200-celA-xg* inoclulant. ^3^ D, effect of ensiling days; T, effect of treatment; D ∗ T, interaction between ensiling days and treatment.

**Table 6 animals-12-01555-t006:** Changes in chemical compositions during the ensiling of low-moisture content *Stylosanthes* silages.

Item ^1^	Treatment ^2^	Days of Ensiling					SEM	*p*-Value ^3^		
		1	3	7	15	45		D	T	D ∗ T
Chemical Compositions										
Dry matter (g kg^−1^ FM)	Control	409.34 ^B^	412.52 ^aB^	412.94 ^B^	408.18 ^B^	433.86 ^aA^	3.17	<0.001	<0.001	<0.001
	LP	412.03 ^A^	416.03 ^aA^	405.66 ^B^	405.52 ^B^	417.12 ^bA^				
	xg	398.04 ^B^	405.62 ^abB^	403.23 ^B^	405.87 ^B^	416.43 ^bA^				
	LP + xg	397.89 ^C^	400.52 ^bC^	402.47 ^BC^	410.62 ^B^	421.24 ^bA^				
Crude protein (g kg^−1^ DM)	Control	161.93	163.28	159.92	161.91	164.13	0.87	0.624	0.272	0.109
	LP	162.31 ^AB^	158.73 ^B^	159.71 ^AB^	163.43 ^A^	161.74 ^AB^				
	xg	163.73	159.37	160.53	162.72	160.21				
	LP + xg	162.41 ^AB^	161.72 ^AB^	159.04 ^B^	159.67 ^AB^	164.18 ^A^				
Water soluble carbohydrate (g kg^−1^ DM)	Control	24.13 ^A^	19.82 ^cB^	16.47 ^cC^	14.81 ^dD^	13.13 ^dE^	0.72	<0.001	<0.001	<0.001
	LP	23.82 ^A^	21.44 ^aB^	18.62 ^bC^	15.92 ^cD^	14.12 ^cE^				
	xg	23.91 ^A^	21.51 ^aB^	20.82 ^aC^	19.33 ^aD^	17.43 ^aE^				
	LP + xg	23.93 ^A^	20.12 ^bB^	18.77 ^bC^	18.21 ^bD^	16.71 ^bE^				
Neutral detergent fiber (g kg^−1^ DM)	Control	595.41 ^A^	583.49 ^aB^	575.43 ^aC^	573.54 ^aCD^	567.72 ^aD^	4.69	<0.001	<0.001	<0.001
	LP	592.02 ^A^	584.83 ^aB^	574.11 ^aC^	564.33 ^bD^	565.43 ^aD^				
	xg	591.48 ^A^	567.12 ^cB^	522.82 ^cC^	494.55 ^dD^	435.48 ^cE^				
	LP + xg	593.03 ^A^	574.22 ^bB^	536.24 ^bC^	511.93 ^cD^	460.22 ^bE^				
Acid detergent fiber (g kg^−1^ DM)	Control	374.11 ^A^	376.76 ^aA^	371.72 ^bAB^	365.21 ^aC^	366.51 ^aBC^	3.93	<0.001	<0.001	<0.001
	LP	375.53 ^AB^	364.61 ^bB^	385.86 ^aA^	371.86 ^aAB^	371.31 ^aAB^				
	xg	372.78 ^B^	375.04 ^aA^	336.12 ^cC^	296.21 ^cD^	297.83 ^bD^				
	LP + xg	375.43 ^A^	366.62 ^bA^	333.57 ^cB^	315.52 ^bC^	288.91 ^bD^				

Means within the same row (A–E) or within the same column (a–d) with different superscripts differ significantly from each other (*p* < 0.05). SEM, standard error of the mean. ^1^ FM, fresh matter. ^2^ LP, *Lactobacillus plantarum* inoculant; xg, *LP-p118200-celA-xg* inoclulant. ^3^ D, effect of ensiling days; T, effect of treatment; D ∗ T, interaction between ensiling days and treatment.

**Table 7 animals-12-01555-t007:** Effect of different study factors and their interaction on chemical compositions of *Stylosanthes* silage.

	Factor	Dry Matter (g kg^−1^ FM)	Crude Protein (g kg^−1^ DM)	Water Soluble Carbohydrate (g kg^−1^ DM)	Neutral Detergent fiber (g kg^−1^ DM)	Acid Detergent Fiber (g kg^−1^ DM)
*p*-value	M	<0.001	<0.001	<0.001	<0.001	<0.001
	T	<0.001	0.797	<0.001	<0.001	<0.001
	D	<0.001	0.029	<0.001	<0.001	<0.001
	M ∗ T	<0.001	0.002	0.002	<0.001	<0.001
	M ∗ D	<0.001	<0.001	<0.001	<0.001	<0.001
	T ∗ D	0.014	0.001	<0.001	<0.001	<0.001
	M ∗ T ∗ D	<0.001	0.004	<0.001	<0.001	<0.001

**Table 8 animals-12-01555-t008:** In vitro dry matter digestibility of *Stylosanthes* silage.

Item	DM	Treatment				SEM	*p*-Value		
		Control	LP	xg	LP + xg		M	T	M ∗ T
48 h in vitro dry matter digestibility (% DM)	28.26%	59.78 ± 0.02 D	61.18 ± 0.04 C	68.57 ± 0.11 A	65.81 ± 0.02 B	0.02	0.505	<0.001	0.24
	39.71%	60.89 ± 0.01 C	61.55 ± 0.06 C	67.59 ± 0.04 A	64.44 ± 0.03 B.				

Note: Different uppercase letters in the same row indicate significant differences (*p* < 0.05).

**Table 9 animals-12-01555-t009:** In vitro neutral detergent fiber digestibility of *Stylosanthes* silage.

Item	DM	Treatment				SEM	*p*-Value		
		Control	LP	xg	LP + xg		M	T	M ∗ T
48 h in vitro neutral detergent fiber digestibility (% DM)	28.26%	27.49 ± 0.03 D	29.13 ± 0.02 C	38.93 ± 0.13 A	37.04 ± 0.06 B	0.04	0.151	<0.001	0.55
	39.71%	27.01 ± 0.02 D	28.22 ± 0.03 C	38.31 ± 0.03 A	36.95 ± 0.01 B.				

Note: Different uppercase letters in the same row indicate significant differences (*p* < 0.05).

**Table 10 animals-12-01555-t010:** Effects of moisture content and additives on in vitro gas production of *Stylosanthes* silage.

DM%		2 h	4 h	6 h	12 h	24 h	36 h	48 h
28.26	CK	5.26	8.75 b	13.76 c	30.62 c	41.23 d	49.41 d	51.31 d
	LP	5.42	8.91 b	14.08 c	31.35 c	42.44 c	50.51 c	52.3 c
	xg	5.32	11.05 a	16.32 a	34.29 a	44.81 a	53.93 a	54.83 a
	LP + xg	5.26	11.53 a	15.18 b	32.94 b	44.34 b	51.89 b	53.68 b
39.71	CK	4.9	8.2 c	14.09 b	31.2 d	38.54 c	47.74 d	50.47 d
	LP	5.22	8.86 c	14.57 b	32.15 c	39.43 c	49.19 c	51.25 c
	xg	5.22	10.86 a	15.57 a	34.42 a	44.16 a	53.04 a	54.36 a
	LP + xg	5.31	10.61 b	14.67 b	33.45 b	42.73 b	51.19 b	53.23 b
*p*-value	M	0.53	0.69	0.51	0.63	0.85	0.28	0.72
	T	<0.001	<0.001	<0.001	<0.001	<0.001	<0.001	<0.001
	M ∗ T	0.41	0.11	0.24	0.33	0.38	0.12	0.16
	SEM	0.02	0.03	0.03	0.02	0.01	0.02	0.02

Note: Different lowercase letters in the same column indicate significant differences (*p* < 0.05).

## Data Availability

Study did not report any data.

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
