# Peer review of "Effects of Different Moisture Levels and Additives on the Ensiling Characteristics and In Vitro Digestibility of Stylosanthes Silage"

_animals, 2022, doi:10.3390/ani12121555_

Round 1

Reviewer 1 Report

Dear editor

My decision: MAJOR revision,

After revision, I want to see it again.

REGARDS

Reviewer 2 Report

Dear Authors,

The reviewed manuscript „Effects of different moisture levels and additives on the ensiling characteristics and in vitro digestibility of Stylosanthes silage” contains the results of an interestingly planned experiment to investigate the effect of different inoculants containing Lactobacillus plantarum strains and two moisture levels of ensilaged material on fermentation quality and in vitro degradability of Stylosanthes silage. Both the introduction to the research problem and the project assumptions, and the methodology were presented in a clear and essential manner, sufficient for the needs of the article submitted for review. The results obtained can be used for elaboration of theoretical basis and technical support for the utilisation of warm-season legume plants for silage production. However, I have few minor comments on it that should be included to improve its clarity.

L 14-17. Both sentences should be moved to the beginning of the abstract.

L 18. The keywords should be different from the words in the title. I suggest changing them.

Introduction: the research hypothesis is missing.

L 91. Alfalfa silage? Please corect it.

Results: Please check on the values for fermentation parameters and chemical composition and units. Currently the values appear to be expressed in % and not in g/kg DM. For example, the dry matter content should be 295.9 g/kg FM and not 29.59 (see table 5). The same comment applies to the values in the other tables.

Conclusions: Please provide information on the practical use of research results and the need of future study.

Author Response

Response to Reviewer 2 Comments

Point 1: L 14-17. Both sentences should be moved to the beginning of the abstract.

Response 1: Correction accepted, two sentences L 14-17 have been moved to the beginning of the abstract.

Point 2: L 18. The keywords should be different from the words in the title. I suggest changing them.

Response 2: Correction accepted, four keywords have been changed to “Anaerobic fermentation, feed stuff, transgenic engineered lactic acid bacteria, fermentation quality ”.

Point 3: Introduction: the research hypothesis is missing.

Response 3: Correction accepted, the research hypothesis has been added after the third paragraph in the Introduction

Point 4: L 91. Alfalfa silage? Please correct it.

Response 4: Due to personal error, Write the wrong material name, "alfalfa silage" in L 91 has been corrected to "Stylosanthes silage".

Point 5: Results: Please check on the values for fermentation parameters and chemical composition and units. Currently the values appear to be expressed in % and not in g/kg DM. For example, the dry matter content should be 295.9 g/kg FM and not 29.59 (see table 5). The same comment applies to the values in the other tables.

Response 5: Correction accepted.(see table 1、2、4、5 and 6)

Point 6: Conclusions: Please provide information on the practical use of research results and the need of future study.

Response 6: Correction accepted, the “Conclusions” have been supplemented with information on the practical use of research results and the need of future study. (see 5. Conclusions)

Round 2

Reviewer 1 Report

     I read revision article carefully

    All recommendations were done.

    My decision: ACCEPT

    REGARDS